# A Temperature Independent Inclinometer Based on a Tapered Fiber Bragg Grating in a Fiber Ring Laser

**DOI:** 10.3390/s21092892

**Published:** 2021-04-21

**Authors:** Weihao Lin, Shengjie Zhou, Liyang Shao, Mang I. Vai, Perry-Ping Shum, Weijie Xu, Fang Zhao, Feihong Yu, Yibin Liu, Yuhui Liu, Shuaiqi Liu

**Affiliations:** 1Department of Electrical and Electronic Engineering, Southern University of Science and Technology, Shenzhen 518055, China; 11510630@mail.sustech.edu.cn (W.L.); 11812717@mail.sustech.edu.cn (S.Z.); shenp@sustech.edu.cn (P.-P.S.); 11930535@mail.sustech.edu.cn (W.X.); 12031197@mail.sustech.edu.cn (F.Z.); 11930480@mail.sustech.edu.cn (F.Y.); 11811808@mail.sustech.edu.cn (Y.L.); 12068026@mail.sustech.edu.cn (Y.L.); 11853004@mail.sustech.edu.cn (S.L.); 2Department of Electrical and Computer Engineering, Faculty of Science and Technology, University of Macau, Macau 999078, China; fstmiv@um.edu.mo

**Keywords:** inclinometer, tapered fiber Bragg grating, fiber ring laser

## Abstract

We demonstrate a new concept for an all-fiber inclinometer based on a tapered fiber Bragg grating (tFBG) in a fiber ring laser (FRL) with the capability of measuring the tilt angle and temperature simultaneously. The sensor performance is analyzed theoretically and investigated experimentally. The dependence of tilt angle on the spectral response in variable temperature conditions was measured. Two inclinometers with different lengths have been fabricated and characterized in FRL. The sensitivity is 0.583 dB/° and 0.849 dB/°, respectively, in the range of 0° to 90°. Thanks to the FRL system, narrow 3-dB bandwidth (<0.1 nm) and high optical signal-to-noise ratio (~60 dB) are achieved. The tFBG in the FRL system can be used for working as a temperature insensitive inclinometer. The results suggested that the proposed inclinometer has the advantages of compact size and convenient manufacture, enhancing its potential for application prospect.

## 1. Introduction

Optical inclinometers have become excellent alternatives in many application fields, including offshore submarine pipeline cable monitoring [1,2], aviation [3], automatic machines [4], human health monitoring [5], etc., owing to their well-known advantages, for instance, immunity to electromagnetic interferences, high sensitivity, and compact size. In the last few decades, various kinds of techniques have been proposed based on metal or semiconductor materials [6] to exploit these inclinometers. However, the structure of the folding pendulum is fragile and not resistant to electromagnetic interference. Therefore, it is necessary to study inclinometers using different technologies to overcome these limitations. Besides, in other applications, for example, virtual reality technology, gesture recognition [7], intelligent machinery [8] and environment monitoring [9] a kind of inclinometer which is cost-effective, extremely accurate, and simple in structure is needed. Over the past few years, fiber-optic inclinometers have been widely investigated because of their inherent advantages, including long-term stability and reusability [10,11,12,13]. Therefore, they perform well under small inclination change conditions. Typically, optical fiber inclinometers consist of a fiber Bragg grating (FBG) [12], tilted fiber grating (TFG) [14] and long period fiber grating (LPFG) [15], etc. Besides, dislocation optical fibers [16] and multi-core optical fibers [17] are also used to manufacture inclinometers.

Liu et al. reported a tapered polymer fiber used as an inclinometer to measure tilt angles [18]. The detection sensitivity is as high as 4.23 dB/°. Despite the high sensitivity of the polymer fiber, the temperature cross sensitivity and the cost of fabrication are fatal drawbacks. Tam et al. proposed an all-fiber two-dimensional inclinometer based on a fiber Bragg grating (FBG) which can simultaneously measure azimuth and inclination [17]. However, its manufacturing process is complicated. Tomasz et al. proposed a new measurement method based on FBG, realizing the detection range of 0°–90° [19]. Nevertheless, the output spectrum is chaotic. Kumar [12] reported an inclinometer base on a pendulum-based structure attached to its cantilever arm with an ultra-high resolution of 0.0008°. However, since, its complex structure the detection range is limited to 0–3°.

Nowadays, optical fiber ring laser sensors have become a hot topic due to their advantages of high signal-to-noise ratio and narrow 3-dB bandwidth [20,21,22,23,24,25,26,27]. Liu et al., proposed a cascaded MZI for strain sensing [28], successfully achieving a sensitivity of 52.5 pm/µε. Martin-Vela et al. reported an inclinometer based on a fiber ring laser [29], where a thin core fiber modal interferometer worked as a filter for the system. Moreover, Fu reported a temperature sensing system based on a fiber ring microwave photonic filter (MPF) [30], which combined microwave photonics with a fiber ring laser. Mateusz proposed a FRL sensor which can simultaneously measure temperature and humidity [31]. Different types of inclinometers and accelerometers based on broadband light sources have been reported [32,33,34], yet to the best of our knowledge, the study of inclinometers based on a tapered fiber Bragg grating in a fiber ring laser has not been proposed.

In this letter, we propose a novel inclinometer based on tFBG in FRL. For the first time the tFBG was utilized in the fiber ring laser sensing system. In the range of 0° to 90°, the tFBG has good temperature response and high sensitivity to inclination variations. Moreover, the change of tilt angle of reflection spectrum under different surrounding temperatures is examined. Furthermore, the elaborate design of the inclinometer in FRL system can choose between two kinds of monitoring instruments according to the specific application requirements. For the first method, the optical spectral analyzer is selected. The tilt angle is measured by detecting the change of light intensity, and the temperature is detected by the spectrum shift. The optical power meter is regarded as an inexpensive measuring tool to measure the tilt angle. It detects the total reflected power to judge the tilt angle without temperature interference. The experimental results show that the designed tFBG has a good response to the tilt angle. The sensitivity of the inclinometers is 0.583 dB/° and 0.849 dB/° in the range of 0–90°. Moreover, benefitting from the single peak output of the laser, the signal to noise ratio is as high as 60 dB and the linewidth is less than 0.1 nm. Furthermore, the thinned central core is essential in the inclinometer for temperature detection. The proposed inclinometer has merits of high detection sensitivity, simple fabrication and low cost.

## 2. Sensor Setup and Principle

As shown in Figure 1. The designed inclinometer is consist of a tFBG stretched in one region. A commercial taper machine (6000 LE-H, Coupler Technology Co., LTD Shandong, China) was selected to fabricate the sensor. The filter of the presented inclinometer is actuality a section of TFBG written on head face of the fiber. Then, a 15 mm long fiber Bragg grating is written into a common single-mode fiber by using scanning phase mask and a continuous ultraviolet laser beam [35,36,37]. The production process includes two steps. The fiber stretching process consists of two steps. Firstly, the end of FBG is placed on the taper machine, and the nozzle is ignited to preheat the fiber. Secondly, the two electric displacement platforms are moved backward at a uniform speed, and the flame melts the fiber for taper. Symmetrical structures with total length of 4.5 and 3.7 mm and diameter of 75 and 86 μm were obtained. In order to improve the photosensitivity of FBG, the tFBG cross section was hydrogasified at 120 bars for one week at 26 °C. As shown in Figure 1. The fiber is tapered at the end of the FBG, leading to the spectral change. Thus, it is convenient to measure the tilt angle and detect the temperature. In the process of taper drawing, the taper structure and length of FBG are monitored in real time. As shown in Figure 2 and Figure 3, the three-port optical circulator for connecting TFBG, light source and detector is optimized. The tilt angle is determined by analyzing the intensity change of OSA or optical power meter.

The working principle of the inclinometer is based on two phenomena: bending loss and selective reflection. Light from a laser source enters the fiber and then passes through the tFBG. When the fiber is bent, the mode field moves forward to one side of the tapered fiber. Therefore, a part of the optical power is coupled to the cladding region and finally dissipated. The rest of the radiation is shifted into the tFBG for selective reflection. Then it goes through the bend area again and is partially lost again. Finally, the output power spectrum is measured by a OSA or optical power meter. The light out of the wavelength range of the FBG passes through the grating and occurs the non-reflective terminal. The main advantage of tFBG is that based on the reflective optical system, the influence of optical power loss will appear twice, thus improving the sensitivity of the inclinometer. More importantly, the temperature is measured as close to the bending region as possible. Benefits from the FRL system, a narrow 3-dB bandwidth (<0.1 nm) and a high optical signal-to-noise ratio (~60 dB) are obtained.

There are three possible transmission paths for incident light coupling into a tapered fiber. In the first optical path, the core wavelength of reflected light must satisfy the FBG filtering condition. The intensity of light decays twice through the taper region. The second optical path is that some cladding modes are re-coupled with the core in the taper region. The strength loss is transferred by the amplitudes of the core and cladding modes compensated in the taper region. The third path is in the core region. The incident light is coupled from the core to the cladding, then re-coupled to the core through the grating. The intensity of light attenuates in core mode.

In this work, the taper profile chosen is a balance between guaranteeing the adiabatic conditions and providing a relatively high bending sensitivity for the inclinometer being analyzed. The power of the lower mode will be transferred to the higher mode if the radius of the tapered fiber changes too fast along the length direction. Therefore, the waist diameter is small enough, the taper is no longer adiabatic. Besides, for ensuring reliable temperature measurement near the bending area, some restrictions are imposed on the maximum length of taper and the position of tFBG. On the other hand, the bending loss increases with the decrease of waist diameter. As shown in Figure 2, tapered fibers with 86 μm and 75 μm waist is selected.

The bending loss per unit length of step index tFBG can be expressed as follows [12]:(1)$$\alpha =\frac{1}{2}{\left(\frac{\pi }{aW^{3}}\right)}^{\frac{1}{2}}{\left(\frac{U}{VK_{1}W}\right)}^{2}\exp\left(-D\cdot R\right)\cdot R^{-\frac{1}{2}}$$
(2)$$D=\frac{4∆W^{3}}{3aV^{2}}$$
where *α*, *R* and Δ are the core radius, bending radius, and core-cladding refractive index difference, respectively. *K*_1_ is a modified Hankel function. *U* and *W* are the transverse propagation constants in the core and cladding. *V* is the normalized frequency.

To ensure reliable temperature sensing in the tFBG region, the maximum length and position of the taper are limited. In this work, we choose to taper at the end of FBG. According to the model provided by a Marcuse experiment [38,39], the theory that the bending loss increases with the decrease of waist diameter is obtained.

A pre-experiment was set up to demonstrate the feasibility of the experimental scheme. As shown in Figure 3 the tFBG was linked with a circulator. Light from broadband light source transmitted to the circulator and reflected when pass through tFBG. The loss of intensity of light dependences on the tilt angle which could be further analyzed in OSA.

The experimental setup of the proposed inclinometer sensor is illustrated in Figure 4. A wavelength division multiplexer (WDM) is used to couple the laser light source in a 1.5 m-long Er-doped fiber (EDF) that works as the gain medium. Besides, an isolator (ISO) is connected to prevent backscattering light. A polarization controller (PC) is embedded in the FRL sensor to regulate the polarization state. 1% port of output coupler is used to extract the light to OSA or optical power meter from the ring cavity for further analysis.

In conclusion, in order to ensure that the proposed structure has enough performance, we choose the parameters of TFBG through repeated experiments. Another way to measure the power spectrum of reflected light is to use an optical power meter instead of OSA. This can effectively reduce the cost of measurement.

When the inclinometer bends, the reflection attenuates strongly at the waist of the tFBG. The bend leaks the laser intensity. The relationship between normalized reflectance coefficient (H) and β can be simply expressed as follows [18]:(3)$$H=pe^{-q\beta^{2}}$$

In Equation (3), *p* and *q* are constants of the exponential fitting curve related to the sensitivity of the inclinometer, and the value of *p* can be equivalent to the total response in the whole measurement range. *q* is the attenuation coefficient of the Gaussian curve in Equation (3). According to Equation (3), θ can be easily determined as follows [18]:(4)$$\beta =\sqrt{\frac{1}{q}\ln\left(\frac{H}{p}\right)}$$

In the view of Equation (2), the θ can be estimated when the values of R are measured. The sensitivity of Equation (1) with respect to θ can be obtained by differentiation as follows [18]:(5)$$S=\frac{\left|dH\right|}{d\theta }$$

As shown in Figure 5, the rotation platform is designed to detect the angle. The rotation angle is adjusted by rotating the nut. By monitoring the light intensity change of the optical power meter, real-time angle monitoring is realized.

## 3. Experimental Results

As shown in Figure 3, pre-experimental results of the inclinometer working in broadband light source is observed. A supercontinuum light source, (ASE-C-N, Hoyatek, Shenzhen, China), an optical spectrum analyzer (OSA, AQ6370D, Yokogawa, Japan) and a circulator are included. The three-port optical circulator works in the wavelength range of 1550 ± 20 nm, which provides the manipulation of the reflection mode and the direction of the light in tFBG. The termination of the FBG is used as the reflector decreasing the retroreflections. The power and intensity fluctuation of reflected light are as low as ±0.009 db/mi and ±0.005 db/min, which indicates that the tilt measurement has good stability. Besides, the optical power meter can be used to replace the OSA to cut down the system cost. As shown in Figure 6, the curvature is measured by optical power meter and spectrometer, respectively. The experimental results show consistency. The black and red lines in Figure 6 almost coincide. Therefore, the optical power meter is a reliable and effective cheap detection instrument.

The reflected light intensity of the inclinometer in broadband light source under different angles is analyzed at the range of 0° to 40° by a Yokogawa AQ6370D spectrometer (wavelength resolution is 0.02 nm). The intensity of reflected light varies with the tilt angle as shown in Figure 7. With the increase of tilt angle, the light intensity enters the cladding mode through the core mode of tFBG, resulting in partial energy loss. To quantify this phenomenon, the relationship between inclination angle and the reflectively is plotted in Figure 8. A consistent linear relationship exists. As the inclination angle e increases, the reflectively decreases, which is consistent with our previous analysis. The sensitivity of the sensor is 0.495 dBm/° and the R^2^ of linear fitting is 0.991. 

The inclinometer is accurately installed on the displacement platform. This means that the angle will be mechanically controlled by the turntable. The spectrum information can be demonstrated specifically by controlling the rotation of the mechanical platform. Firstly, the rotation angle of the turntable is controlled in the pre-experiment. For observing the change of light intensity more accurately, the rotation step is 10°. In practical application the tFBG sensor needs to be packaged to improve the stability. Nevertheless, the spectrum of the broadband light source measuring the inclination angle is disturbed, which may lead to the inaccuracy of the selected data, as shown in Figure 7. Moreover, the low detection sensitivity may affect the resolution of the inclinometer and increase the error, resulting in the decrease of usability.

The performance of inclinometer in FRL system is verified immediately, as shown in Figure 9. At first, tFBG with waist of 86 μm and the distance from bending area to FBG center of 1.9 mm was selected as inclinometer. The light intensity decreases with the increase of inclination angle. As shown in Figure 10 and Figure 11, a stable linear relationship is maintained in the measurement range of 0~80°. The sensitivity of the inclinometer is between −0.849 dBm/°and −0.868 dBm/°. The R^2^ of linear fitting is 0.992, as shown in Figure 10. Compared with the broadband light source, the sensitivity is further improved because of the more concentrated light intensity distribution in the laser cavity. As shown in Figure 11, for the sake of further demonstrated the resolution of inclinometer in FRL system. The experimental results show that the sensitivity of the sensor is 0.849 dbm/° with detection step of 3°, which is consistent with the result of 0.868 dbm/° as shown in Figure 10. The R^2^ of linear fitting is 0.999, which verifies the reliability and stability of the system. The result show that the detection resolution of inclinometer is better than 3°.

As mentioned above, the system can be further used as a temperature sensor. Figure 12 shows the transmission spectra of the tFBG at different temperature with the tapered fiber waist of 86 μm. It can be seen from the Figure 12 that the spectrum moves smoothly without the light intensity change under the range of 0 to 80°. As shown in Figure 13, the temperature sensing characteristics remain consistence when the inclination angle changes in a large range (80°). The temperature sensitivity of the inclinometer as a temperature sensor is 12 pm/°C, which is a representative value of a FBG. The result is affected by the thermal expansion coefficient of the single mode fiber. Besides, the experimental results show that the transformation of light intensity is smaller than 0.05 dBm at the range of 0 °C to 50 °C, which is negligible for an inclinometer.

The relationship between the measured total optical power and the inclination angle of the reflection spectrum is shown in Figure 14, when the tFBG waist becomes 75 μm. Under this condition, care should be taken to minimize any unnecessary back reflections in the system. As shown in Figure 15. The sensitivity is 0.631 dBm/° and R^2^ of linear fitting is 0.991. As shown in Figure 16, we can further explore the detection resolution of the inclinometer. The experimental results show that the sensitivity is 0.631 dBm/° and the R^2^ of the linear fit is 0.994 with the detection step of 3°. The discrepancy may be caused by residual strains and mechanical errors. The experimental results show that the detection resolution of the inclinometer is better than 3°.

When the waist width of TFBG is 75 μm, the FRL system can produce a single longitudinal mode laser output with a bandwidth of 0.04 nm and a signal-to-noise ratio of 50 dB, as shown in Figure 14. Wavelength shift and power fluctuation were also detected. As shown in Figure 17, within 250 min, the laser energy and wavelength fluctuate in the range of ±0.52 dBm/h and ±0.43 nm/h, respectively. The analysis shows that the designed tFBG is easy to be affected by the external stress because it is not packaged. This will lead to the deviation of light intensity stability. To further improve the practical application ability and accuracy of the system. Flexible materials such as PDMS and PMMA can be selected as packaging devices.

## 4. Discussion

The proposed inclinometer based on tFBG in a FRL system was compared with other inclinometers like those described in references [40,41,42,43,44,45,46,47,48]. The inclination angle range is shown in Table 1. The proposed inclinometer is an impactful method to enhance the detection range. The bound of response rate is principal owing to the tapered lengths, and there are some issues could be improved further in the future.

The sensitivity of the inclinometer working with the laser is almost twice that achieved when working with a broadband light source. Besides, the sensitivity can be improved by changing the taper depth and position. In addition, if the waist is too thin, the light intensity will leak out at a very small angle, which impact the detection range. Here, our proposed inclinometer cannot define the direction. The problem could be solved by chosing a tilted fiber grating. Figure 18a shows the tapered FBG with waist equal to 50 µm. However, dust hast a great impact on the experimental results. Therefore, a 86 µm waist is selected.

Regarding repeatability and reproducibility, the inclinometer has been used several times (>10 times) with change in measurands and we find similar outcomes each time. The computed value the error of the measurement is on the order of 10^–3^. The sensors were used for repeated experiments quite a few time (>10 times and >5 h) and the results were similar.

## 5. Conclusions

In conclusion, an inclinometer based on tFBG in a FRL structure is proposed and demonstrated. The inclinometer has the ability to simultaneously monitor inclination and temperature. The sensitivity is 0.849 dBm/° and 0.583 dBm/°, respectively with a detection range from 0° to 90°. For temperature sensing, sensitivity of 12 pm/°C is achieved without fluctuating light intensity. This new concept of an inclinometer based on tFBG in FRL structure provides the following advantages: (a) narrow 3-dB bandwidth (<0.1 nm) and high optical signal-to-noise ratio (~60 dB) are achieved. (b) An increase in the light intensity since the FRL system has better spectral response. (c) Consistency of the temperature response under different inclinometer angles. (d) A low-cost optical power meter can be used to replace OSA for tilt measurements. Moreover, the proposed inclinometer has the advantages of compact size and convenient manufacture, enhancing its application prospects.

## Figures and Tables

**Figure 1 sensors-21-02892-f001:**
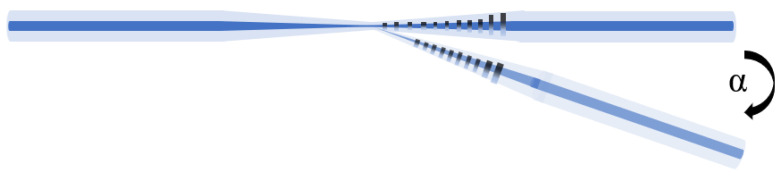
Schematic illustration of the inclinometer.

**Figure 2 sensors-21-02892-f002:**
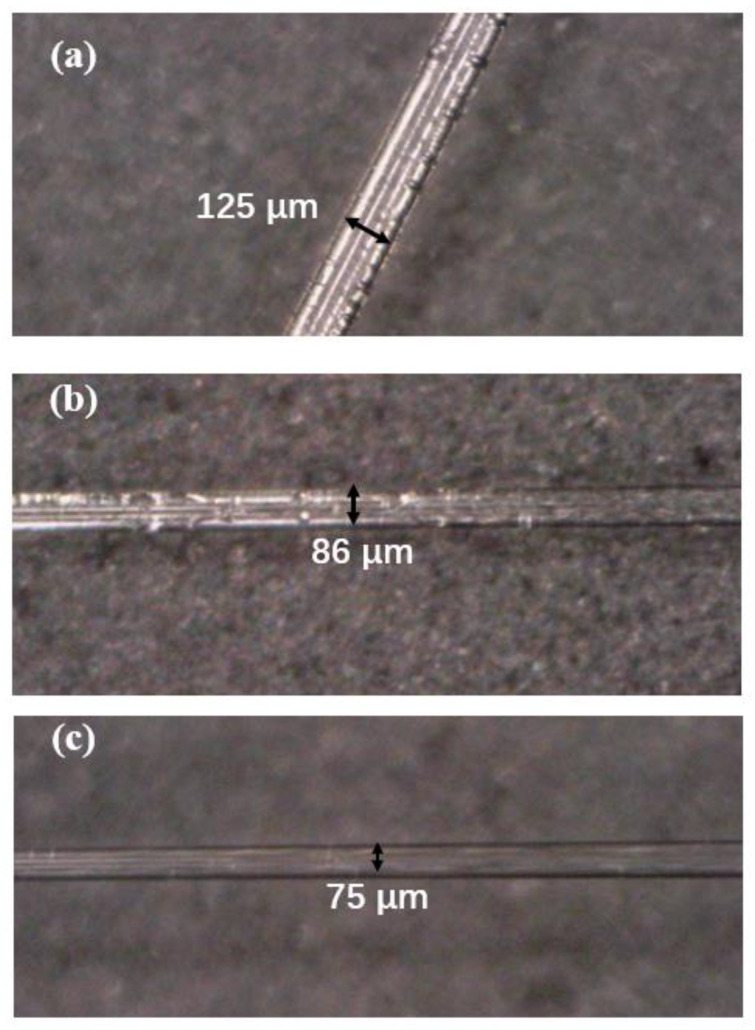
Microphotographs of different structures of the taper-shaped FBG, (**a**) with 125 µm waist (**b**) with 86 µm waist. (**c**) with 75 µm waist.

**Figure 3 sensors-21-02892-f003:**
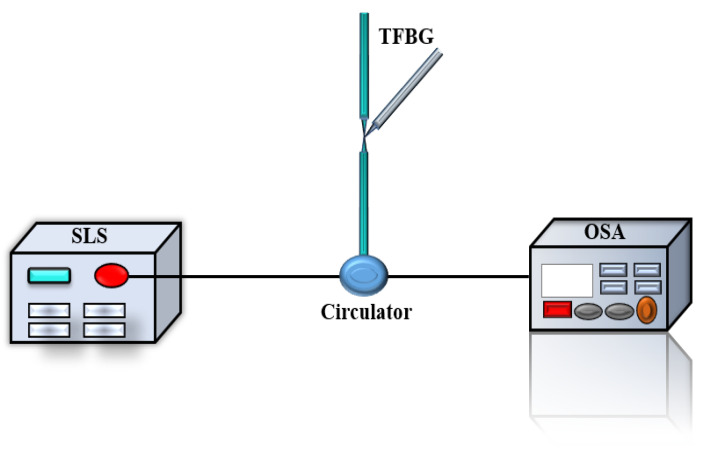
Schematic diagram of the experimental setup for the inclination measurement system.

**Figure 4 sensors-21-02892-f004:**
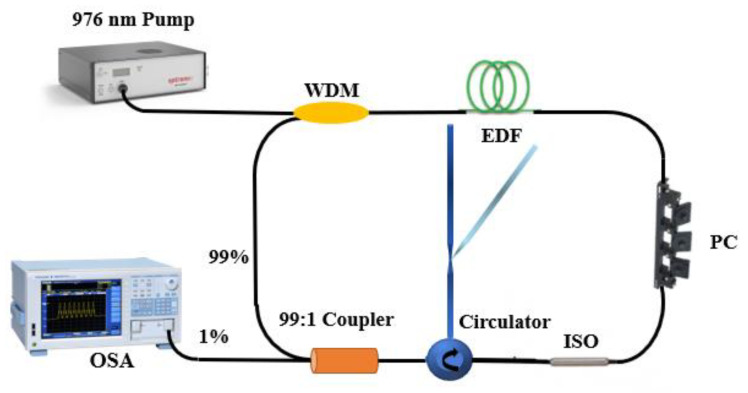
Experiment setup of the proposed fiber laser inclinometer.

**Figure 5 sensors-21-02892-f005:**
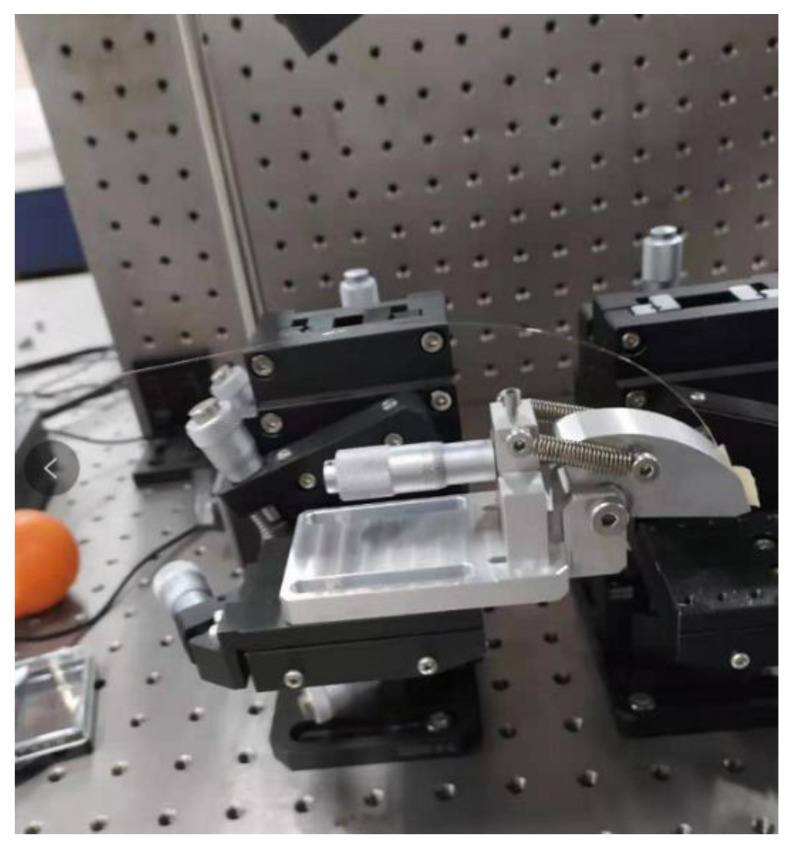
Experimental setup of turntable.

**Figure 6 sensors-21-02892-f006:**
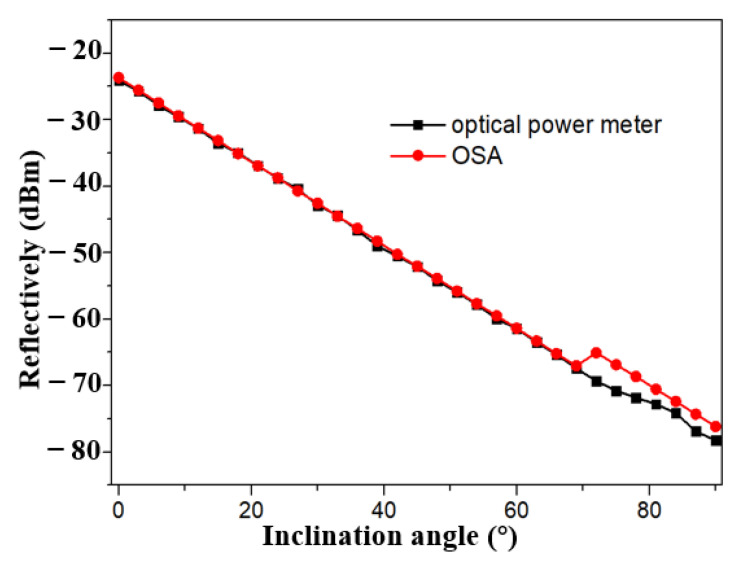
Optical power at peak reflection wavelength for various inclination angle (red line: optical power meter; black line: OSA).

**Figure 7 sensors-21-02892-f007:**
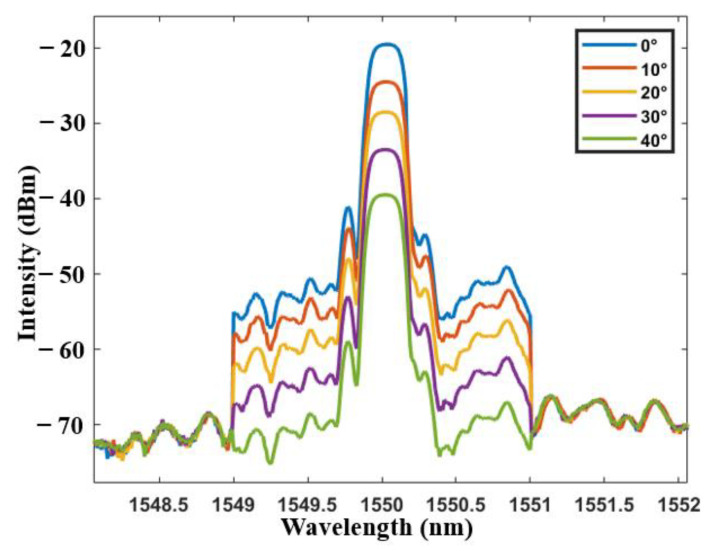
Transmission spectra of the tFBG at different angles.

**Figure 8 sensors-21-02892-f008:**
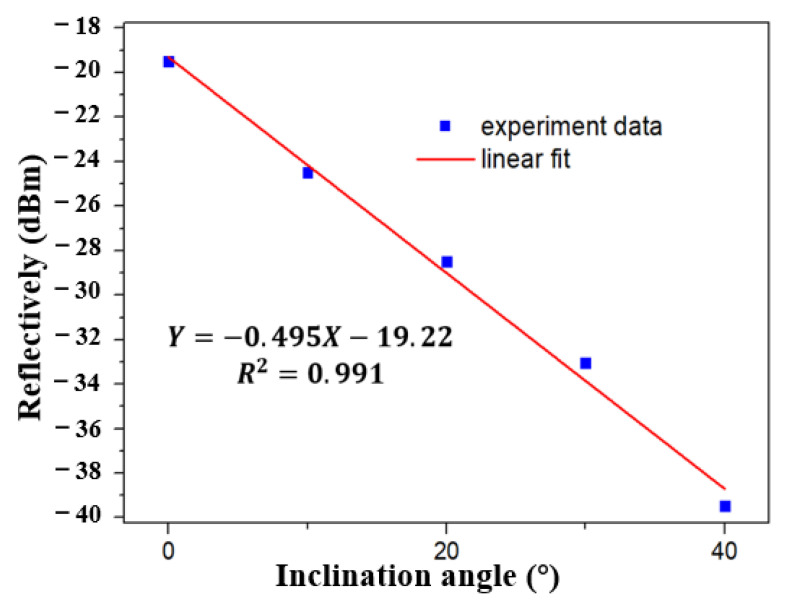
Linear fitting result of the relationship between reflectively and inclination angle.

**Figure 9 sensors-21-02892-f009:**
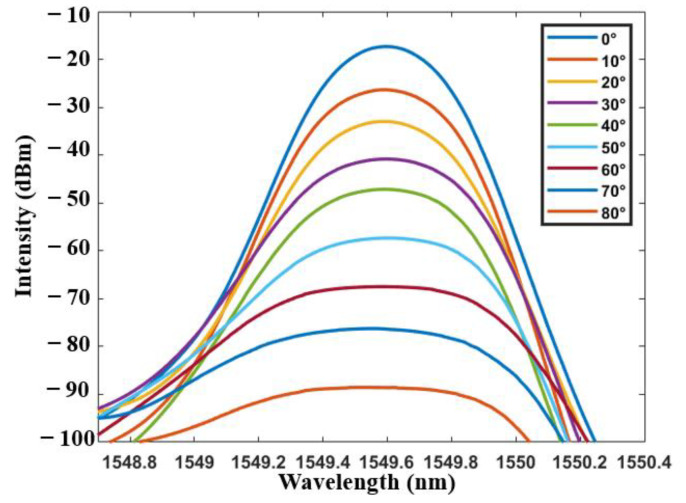
The dependence of the reflection spectrum at the wavelength of the reflection peak at different inclination angles in FRL with waist equals 86 µm.

**Figure 10 sensors-21-02892-f010:**
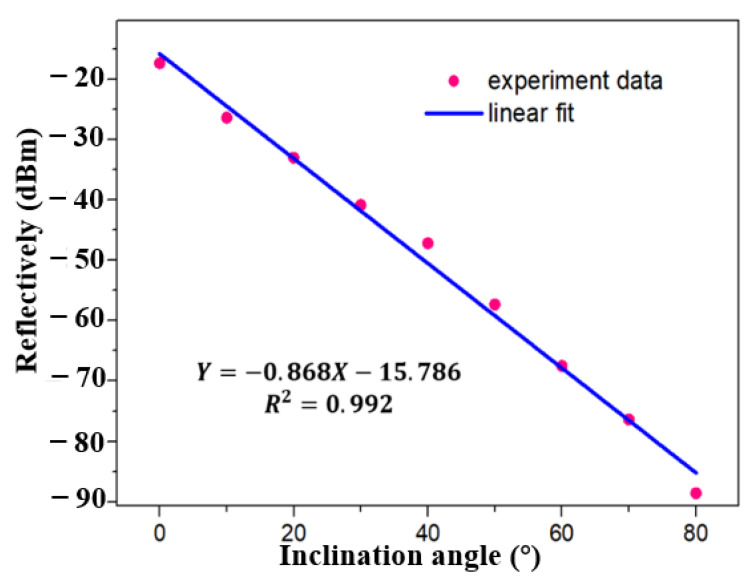
Linear fitting result of the relationship between reflectively and inclination angle in FRL system with step of 10° with waist equals 86 µm.

**Figure 11 sensors-21-02892-f011:**
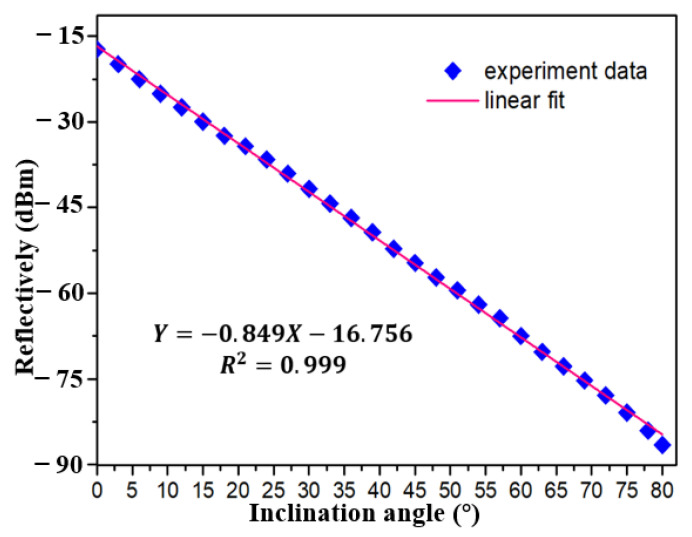
Linear fitting result of the relationship between reflectively and inclination angle in FRL system with step of 3°.

**Figure 12 sensors-21-02892-f012:**
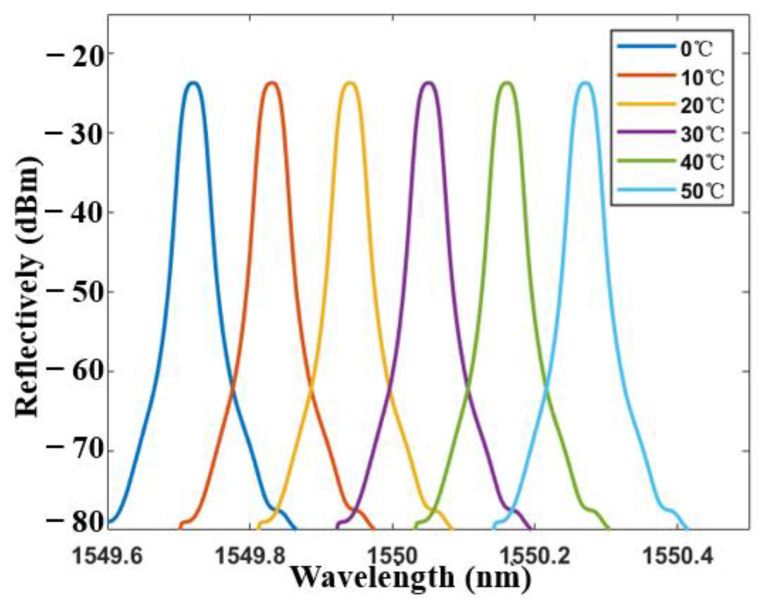
Transmission spectra of the tFBG at different temperature.

**Figure 13 sensors-21-02892-f013:**
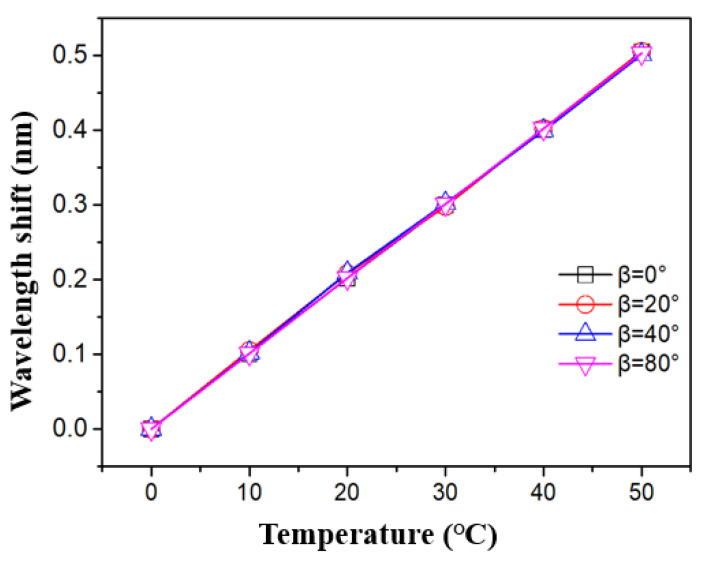
The influence of temperature on a wavelength shift.

**Figure 14 sensors-21-02892-f014:**
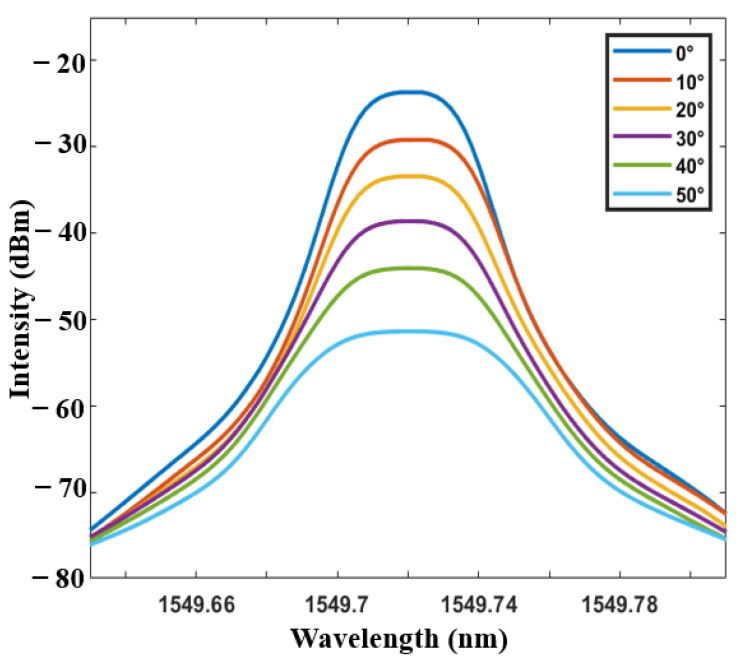
The dependence of the reflection spectrum at the wavelength of the reflection peak at different inclination angles in FRL with waist equals 75 µm.

**Figure 15 sensors-21-02892-f015:**
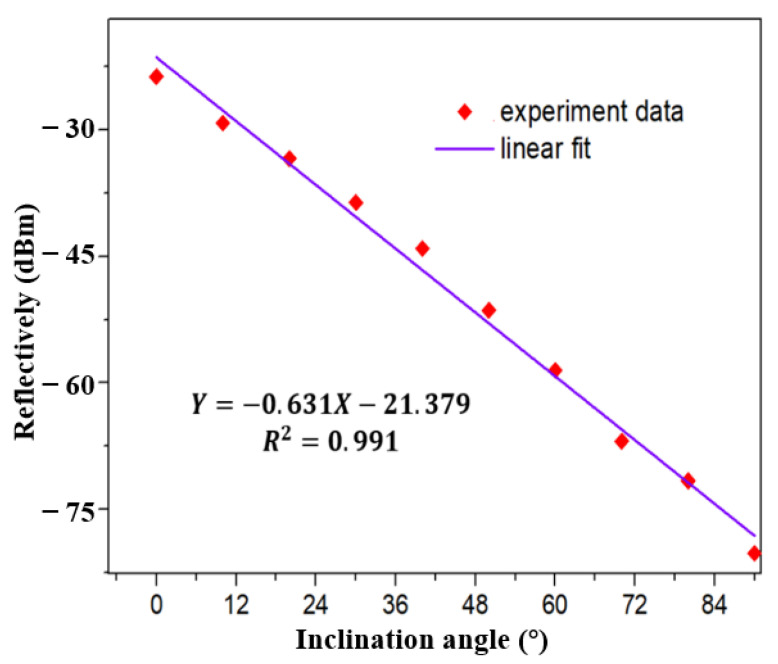
Linear fitting result of the relationship between reflectively and inclination angle in FRL system with step of 10° with waist equals 15 µm.

**Figure 16 sensors-21-02892-f016:**
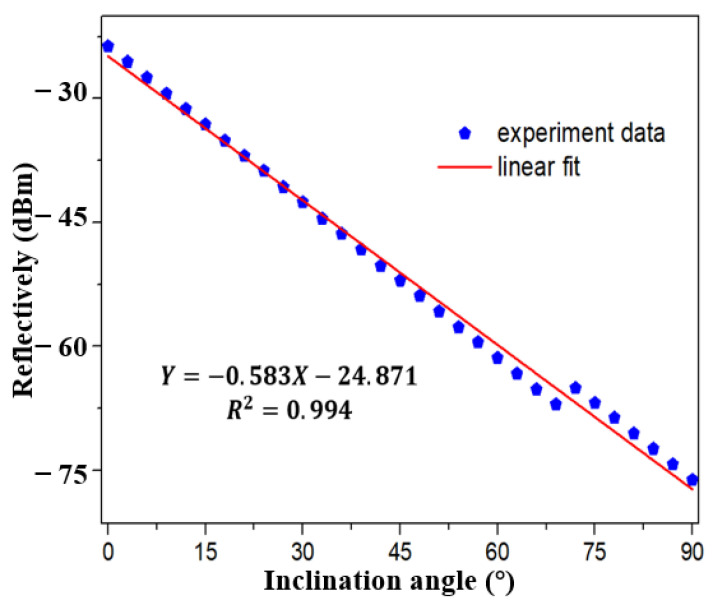
Linear fitting result of the relationship between reflectively and inclination angle in FRL system with step of 3°.

**Figure 17 sensors-21-02892-f017:**
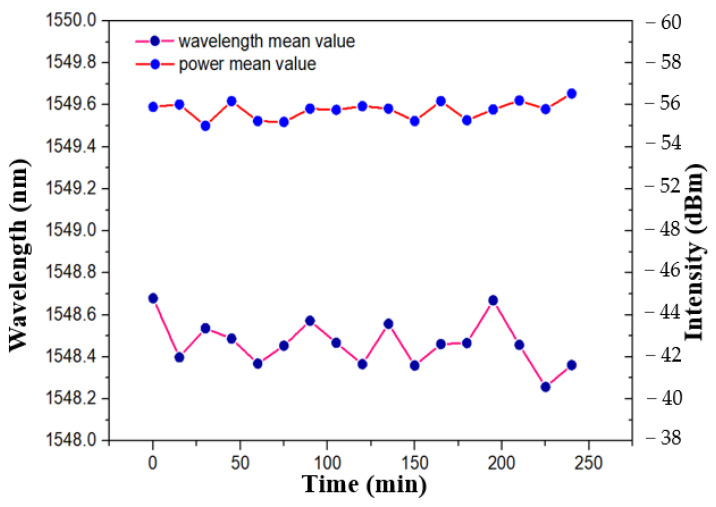
Test for time stability of wavelength shift and power fluctuation (at 51°).

**Figure 18 sensors-21-02892-f018:**
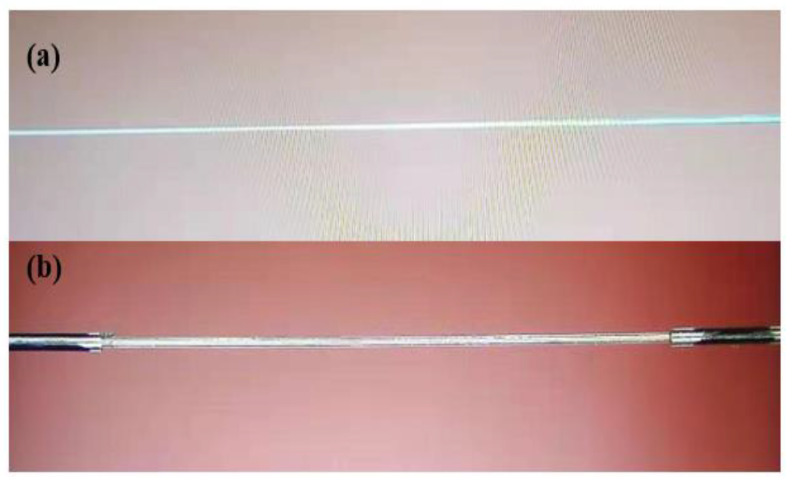
Scheme of the (**a**) FBG with waist of 50 µm (**b**) normal FBG.

**Table 1 sensors-21-02892-t001:** The detection range comparation with other optical fiber inclinometer.

Structures	Tilt Angle	References
Tilted fiber Bragg grating	−12–12°	[40]
Tapered Polymer Fiber	−6–6°	[18]
Peanut-Shape Structure	0–6.66°	[41]
Fiber Bragg Grating	0–1°	[42]
A couple of matched FBGs	−6°	[43]
Cascaded FBG	/	[44]
all-fiber loop mirror	−45–45°	[45]
small size Fiber Bragg Grating	−3–3°	[46]
fiber Bragg grating-based inclinometer	/	[47]
extrinsic Fabry-Perot interferometer	0–8°	[48]
Current work	0–90°	

## Data Availability

Not applicable.

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
