# Peer review of "A Temperature Independent Inclinometer Based on a Tapered Fiber Bragg Grating in a Fiber Ring Laser"

_sensors, 2021, doi:10.3390/s21092892_

Round 1

Reviewer 1 Report

The paper is badly written with too many mistakes. The reader gets disappointed while scrolling through the text:

  • wrong use of punctuation;
  • nouns and verbs confused;
  • ref [30] indicated using Roberto et al – but this is the first name;
  • wrong noun spelling;
  • the term length used instead of diameter;
  • equations using different symbols than those used in the text;
  • from figure 4 the numbering of captions and text does not match;
  • many sentences are not clearly understandable;

The idea is that the paper has not even be read twice before submission.

In addition, a clear explanation of the operating principle is not given. The sensing mechanism, the usage of the two tapers/bragg gratings are not clearly described.

The usage of the rotating plate is not clearly described.

The text describing figure 5 is reporting data the range of 0° to 60° but the figure reports the range of 0° to 40°.

Figure 10 can be avoided.

Label in figure 14 does not discriminate the curves

In conclusion, the sentence “… and there are some issues could be improved further in the future.” Is questionable, what does it mean? Which issues are the authors talking about?

Also, the authors say “needed a cost-effective, extremely accurate, and simple-structured inclinometer.” But the usage of an optical spectrum analyser is not cost. effective. The work would be much more interesting with a different solution for detecting the signal.

Please, go through the whole manuscript and improve presentation and scientific significance.

Author Response

Dear Professor,

Thank you for your helpful suggestions and comments, which have improved the quality of our manuscript greatly.

Revisions according to comments made by Reviewer 1:

In conclusion, the sentence “… and there are some issues could be improved further in the future.” Is questionable, what does it mean? Which issues are the authors talking about?

Also, the authors say “needed a cost-effective, extremely accurate, and simple-structured inclinometer.” But the usage of an optical spectrum analyser is not cost. effective. The work would be much more interesting with a different solution for detecting the signal.

Please, go through the whole manuscript and improve presentation and scientific significance.

Comment 1. The paper is badly written with too many mistakes. The reader gets disappointed while scrolling through the text:

  • wrong use of punctuation;
  • nouns and verbs confused;
  • ref [30] indicated using Roberto et al – but this is the first name;
  • wrong noun spelling;
  • the term length used instead of diameter;
  • equations using different symbols than those used in the text;
  • from figure 4 the numbering of captions and text does not match;
  • many sentences are not clearly understandable;

Reply: Authors appreciate the concerns of the reviewer and thankful to him for his comments. This total paper has been reviewed and corrected accordingly. Many lines are corrected and changed for ease of understanding  

Thanks for the staff’s checking and the mistake of the text format has been corrected throughout the article.

Comment 2. “In addition, a clear explanation of the operating principle is not given. The sensing mechanism, the usage of the two tapers/bragg gratings are not clearly described.”

Reply: This is a really a thoughtful comment from the reviewer, we appreciate the reviewer comment.

Here, we add more analysis to explain the sensing mechanism and the usage of the two tapers/Bragg gratings. A mechanism principle of the sensors is given in the underline part.

Theoretical part:

The filter of the presented inclinometer is actuality a section of TFBG written on head face of the fiber. By heating and stretching, the taper was made on the standard single-mode matched FBG. Symmetrical structures with total length of 3.57mm and 4.78mm and waist diameter of 75 and 86 μm were obtained. In order to improve the photosensitivity of FBG, the TFBG cross section was hydrogenated at 120 bars for one week at room temperature. The growth and length of the TFBG reflectance spectrum were continuously monitored during the taper process.

The principle of operation of the presented inclinometer is based on two phenomena: bending losses and selective reflection. Light from a broadband optical source is launched into the optical fiber and then travels through the tapered section. When the fiber is bent, the mode field is shifted towards one side of the tapered fiber. Thus, a part of the optical power is coupled into the cladding region and finally partially dissipates. The remaining part of the radiation is selectively reflected by the fiber Bragg grating written into the transition of the taper. Then it travels again through the bending area and once again is partially lost. Finally, the output power spectrum is measured by an optical spectrum analyzer. The light outside the wavelength range of the grating passes through it towards the reflection-free termination. The main advantage of the TFBG application, as in most reflection based optical systems, is that the effect of the optical power loss occurs twice, which enhances inclinometer sensitivity. Moreover, the grating written in the taper transition ensures that the temperature can be simultaneously measured as close to the bending region as possible.

In this work, the taper profile chosen was a trade-off between ensuring the adiabaticity criterion and providing relatively high bending sensitivity of the analyzed inclinometer. If the radius of the tapered optical fiber changed too rapidly along its length, it would lead to shifting of the power of the LP01 mode to the higher order modes. Thus, when the waist diameter is too small, the taper is no longer adiabatic and becomes lossy. Moreover, to ensure reliable temperature measurements close to the bending region, some constraints on the maximum length of the taper and TFBG location are imposed. On the other hand, bending losses at a given inclination angle (and thus sensor sensitivity) increase with the reduction of the taper waist diameter. This may be concluded from the model presented by Marcuse  or from experimental data provided by Kim et al. It means that in contrast to the adiabaticity criterion, an optical fiber taper with adequately small waist diameter ensures satisfactory bending sensitivity of the inclinometer.

In conclusion, in order to ensure that the proposed configuration has enough performance, we choose the parameters of TFBG according to repeated experiments. Another way to measure the power spectrum of reflected light is to use an optical power meter instead of OSA. This can effectively reduce the cost of measurement.

Ref:

[1] K. JeË›drzejewski, “Biconical fused taper—A universal fibre devices technology,” Opt-Electron. Rev., vol. 8, no. 2, pp. 153–159, 2000.

[2] T. Osuch, K. JeË›drzejewski, L. Lewandowski, and W. Jasiewicz, “Shaping the spectral characteristics of fiber Bragg gratings written in optical fiber taper using phase mask method,” Photon. Lett. Poland, vol. 4, no. 4, pp. 128–130, Dec. 2012.

[3] T. A. Birks and Y. W. Li, “The shape of fiber tapers,” J. Lightw. Technol.,vol. 10, no. 4, pp. 432–438, Apr. 1992.

[4] D. Marcuse, “Curvature loss formula for optical fibers,” J. Opt. Soc. Amer., vol. 66, no. 3, pp. 216–220, Mar. 1976.

[5] K.-T. Kim, J.-H. Kang, H. Seung, and K. Im, “In-line variable optical attenuator based on the bending of the tapered single mode fiber,” J. Opt. Soc. Korea, vol. 13, no. 3, pp. 349–353, Sep. 2009.

Comment 3. “The usage of the rotating plate is not clearly described.”

      Reply: Authors thank for the reviewer comments; it is very significant.

As shown in the figure, we use the designed rotating platform to detect the rotation angle. Adjust the rotation angle by rotating the nut. Real time angle monitoring is realized by monitoring the intensity change of spectrometer.

Fig.1. Schematic diagram of rotating plate device

Comment 4. “The text describing figure 5 is reporting data the range of 0° to 60° but the figure reports the range of 0° to 40°?”

Reply: We appreciate the observations of the reviewer. In this paper our aim is just to show the possibility of the FRL sensor as a meaningful curvature sensing application. The schematic diagram is only used to show that the sensor can be used for different curvature sensing applications under the condition of broadband light source, so we do not fully show the test results. Here, we are very grateful to the reviewers for their suggestions and unified the schematic diagram.

Fig.2. optical power at peak reflection wavelength for various inclination angle

Comment 5. “Figure 10 can be avoided?”

Reply: Authors thank for the reviewer comments; it is very significant. Therefore, we deleted Figure 10.

Comment 6. “Label in figure 14 does not discriminate the curves?” Also, the authors say “needed a cost-effective, extremely accurate, and simple-structured inclinometer.” But the usage of an optical spectrum analyser is not cost. effective. The work would be much more interesting with a different solution for detecting the signal.

Reply: We thank the reviewer for his remarks, the curves of these figure is 51°. In addition, the spectrometer is the most conventional detection instrument for detecting light intensity. We also choose to use the optical power meter for monitoring, and get the detection results consistent with the spectrometer measurement results. This verifies the stability and repeatability of the sensor. As an alternative instrument of spectrometer, optical power meter can greatly reduce the measurement cost. We further describe the consistency of the two instruments.

Figure 3. Test for stability of wavelength and power.

Figure 4. optical power change of OSA and power meter

We thank the reviewers once again for their helpful suggestions and comments, which have improved the quality of our manuscript greatly.

Yours sincerely,

Li-Yang Shao/ Weihao Lin

Reviewer 2 Report

The paper presents an fiber-optic inclinometer based on the effect of the light loss in bent optical fiber with the embedded Bragg grating (FBG). As the reflection spectrum of FBG depends on the temperature, this sensor can simultaneously used for temperature measurements.

The manuscript, to my best knowledge, presents a new original configuration with this sensor inside the ring laser cavity that demonstrates some interesting results. At the same time, the study of the tapered fiber with embedded FBG does not seem original as similar configurations have been presented in earlier works, see, for example Ji, Chongke, Zhao, Chun-Liu, Kang, Juan, Dong, Xinyong, Jin, Shangzhong: Multiplex and simultaneous measurement of displacement and temperature using tapered fiber and fiber Bragg grating , Review of Scientific Instruments 83(5), 053109, 2012.

For this reason, I think that authors should provide more comprehensive analysis of earlier works to better show the originality of their research.

Besides that, the quality of the presentation must be substantially improved because some statements and descriptions are not clear.

English also needs an essential redaction. The comments regarding the clarity of presentation and English could be found in the attached PDF file of the original manuscript with annotations. The text that requires corrections is marked with color and supplied with the inline notes. Please note, my comments cover only a small part of the text that needs a serious revision.

Author Response

Dear Professor,

Thank you for helpful suggestions and comments, which have improved the quality of our manuscript greatly.

Revisions according to comments made by Reviewer 2:

Comment 1. The paper presents a fiber-optic inclinometer based on the effect of the light loss in bent optical fiber with the embedded Bragg grating (FBG). As the reflection spectrum of FBG depends on the temperature, this sensor can simultaneously use for temperature measurements.

The manuscript, to my best knowledge, presents a new original configuration with this sensor inside the ring laser cavity that demonstrates some interesting results. At the same time, the study of the tapered fiber with embedded FBG does not seem original as similar configurations have been presented in earlier works, see, for example Ji, Chongke, Zhao, Chun-Liu, Kang, Juan, Dong, Xinyong, Jin, Shangzhong: Multiplex and simultaneous measurement of displacement and temperature using tapered fiber and fiber Bragg grating , Review of Scientific Instruments 83(5), 053109, 2012.

For this reason, I think that authors should provide more comprehensive analysis of earlier works to better show the originality of their research.

Reply:  This is a really a thoughtful comment from the reviewer, we appreciate the reviewer comment.

Here, we add more analysis to explain the sensing mechanism and the usage of the two tapers/bragg gratings. A mechanism principle of the sensors is given in the underline part. The references provided are added for comparison also.

Theoretical part:

The filter of the presented inclinometer is actuality a section of TFBG written on head face of the fiber. By heating and stretching, the taper was made on the standard single-mode matched FBG. Symmetrical structures with total length of 45 and 37 mm and waist diameter of 75 and 86 μm were obtained. In order to improve the photosensitivity of FBG, the TFBG cross section was hydrogenated at 120 bars for one week at room temperature. The growth and length of the TFBG reflectance spectrum were continuously monitored during the taper process.

The principle of operation of the presented inclinometer is based on two phenomena: bending losses and selective reflection. Light from a broadband optical source is launched into the optical fiber and then travels through the tapered section. When the fiber is bent, the mode field is shifted towards one side of the tapered fiber. Thus, a part of the optical power is coupled into the cladding region and finally partially dissipates. The remaining part of the radiation is selectively reflected by the fiber Bragg grating written into the transition of the taper. Then it travels again through the bending area and once again is partially lost. Finally, the output power spectrum is measured by an optical spectrum analyzer. The light outside the wavelength range of the grating passes through it towards the reflection-free termination. The main advantage of the TFBG application, as in most reflection based optical systems, is that the effect of the optical power loss occurs twice, which enhances inclinometer sensitivity. Moreover, the grating written in the taper transition ensures that the temperature can be simultaneously measured as close to the bending region as possible.

In this work, the taper profile chosen was a trade-off between ensuring the adiabaticity criterion and providing relatively high bending sensitivity of the analyzed inclinometer. If the radius of the tapered optical fiber changed too rapidly along its length, it would lead to shifting of the power of the LP01 mode to the higher order modes. Thus, when the waist diameter is too small, the taper is no longer adiabatic and becomes lossy. Moreover, to ensure reliable temperature measurements close to the bending region, some constraints on the maximum length of the taper and TFBG location are imposed. On the other hand, bending losses at a given inclination angle (and thus sensor sensitivity) increase with the reduction of the taper waist diameter. This may be concluded from the model presented by Marcuse  or from experimental data provided by Kim et al. It means that in contrast to the adiabaticity criterion, an optical fiber taper with adequately small waist diameter ensures satisfactory bending sensitivity of the inclinometer.

In conclusion, in order to ensure that the proposed configuration has enough performance, we choose the parameters of TFBG according to repeated experiments. Another way to measure the power spectrum of reflected light is to use an optical power meter instead of OSA. This can effectively reduce the cost of measurement.

In addition, it is compared with reference [44]. The proposed method using TFBG in fiber ring laser is more sensitive to curvature. The curvature can be monitored in real time by optical power meter, which greatly reduces the design cost.

Ref:

[1] K. JeË›drzejewski, “Biconical fused taper—A universal fibre devices technology,” Opt-Electron. Rev., vol. 8, no. 2, pp. 153–159, 2000.

[2] T. Osuch, K. JeË›drzejewski, L. Lewandowski, and W. Jasiewicz, “Shaping the spectral characteristics of fiber Bragg gratings written in optical fiber taper using phase mask method,” Photon. Lett. Poland, vol. 4, no. 4, pp. 128–130, Dec. 2012.

[3] T. A. Birks and Y. W. Li, “The shape of fiber tapers,” J. Lightw. Technol.,vol. 10, no. 4, pp. 432–438, Apr. 1992.

[4] D. Marcuse, “Curvature loss formula for optical fibers,” J. Opt. Soc. Amer., vol. 66, no. 3, pp. 216–220, Mar. 1976.

[5] K.-T. Kim, J.-H. Kang, H. Seung, and K. Im, “In-line variable optical attenuator based on the bending of the tapered single mode fiber,” J. Opt. Soc. Korea, vol. 13, no. 3, pp. 349–353, Sep. 2009.

[44] Ji, C.;  Zhao, C. L.;  Kang, J.;  Dong, X.; Jin, S., Multiplex and simultaneous measurement of displacement and temperature using tapered fiber and fiber Bragg grating. Rev Sci Instrum 2012, 83 (5), 053109.

Comment 2. Besides that, the quality of the presentation must be substantially improved because some statements and descriptions are not clear.

English also needs an essential redaction. The comments regarding the clarity of presentation and English could be found in the attached PDF file of the original manuscript with annotations. The text that requires corrections is marked with color and supplied with the inline notes. Please note, my comments cover only a small part of the text that needs a serious revision.

Reply: Authors appreciate the concerns of the reviewer and thankful to him for his comments and really appreciate his help. This total paper has been reviewed and corrected accordingly. Many lines are corrected and changed for ease of understanding.

This is a really a thoughtful comment from the reviewer, we appreciate the reviewer comment.

Here, we add more analysis to explain the sensing mechanism and the usage of the two tapers/bragg gratings. A mechanism principle of the sensors is given in the underline part.

We thank the reviewers once again for their helpful suggestions and comments, which have improved the quality of our manuscript greatly.

Yours sincerely,

Li-Yang Shao/ Weihao Lin

Reviewer 3 Report

The paper presents new concept for an all-fiber inclinometer based on a tapered fiber Bragg grating (tFBG) in fiber ring laser (FRL) with the capability of measuring the tilt angle and temperature, simultaneously. I have some suggestions to improve the paper.

a. Introduction section: it is missing the works related to FBG accelerometers and even FBGs based angle or tilt sensors, which is a large research field and there are many works to be reported. The focus of this paper is inclinometer and there is a lack of works cited about his concern. Please read and consider some references about this point in order to improve the introduction:

FBG acceleremeters: 1. Diaphragm Based Fiber Bragg Grating Acceleration Sensor with Temperature Compensation. Sensors 17, 218, 2017.

FBGs Angle/tilt measurements: 2. A Rotation Independent In-Place Inclinometer/Tilt Sensor Based on Fiber Bragg Grating," IEEE Transactions on Instrumentation and Measurement, vol. 68, no. 8, pp. 2943-2953, Aug. 2019; 3. Polymer optical fiber Bragg gratings in CYTOP fibers for angle measurement with dynamic compensation, Polymers 10 (6), 674, 2018; 4. FPI-POFBG Angular Movement Sensor Inscribed in CYTOP Fibers With Dynamic Angle Compensator, IEEE Sensors Journal 20 (11), 5962-5969, 2020.

b. The TFBG normally is used for Tilted Fiber Bragg Grating (TFBG) and it is not the best acronym for tapered FBG, maybe use tFBG.

c. When the authors talk about the tilt angle and temperature measurements simultaneously, it must be more discussed and add more results because it is not well demonstrated as simultaneously way.

d. More details about FBGs fabrication must be added to the manuscript. Nothing is added about FBG fabrication (please consider to add some literature about FBG fabrication in different materials such as Advances on polymer optical fiber gratings using a KrF pulsed laser system operating at 248 nm, Fibers 6 (1), 13, 2018.

e) Figure 2. (a) Scheme of the normal FBG. (b) Scheme of the tapered FBG (86µm). (c) Scheme of the tapered FBG (75µm). For me, it is not a scheme but a microscope images right?

f. For a commercial application the instrumentation like OSA and SLS could be an issue due to portability and even expensive cost. Please, add some words about that and consider to include some literature about low-cost interrogation systems as a potential solution to decrease costs and increase the portability for a commercial solution. Please read and consider the literature: Fast peak-tracking method for FBG reflection spectrum and nonlinear error compensation," Opt. Lett. 45, 451-454 (2020); Wireless, Portable Fiber Bragg Grating Interrogation System Employing Optical Edge Filter. Sensors 19, 3222, 2019.

g. How about the repeatability of the proposed sensor (if carry out many cycles of tests) and about the reproducibility of the proposed sensor (due to tapering process we can obtain similar probes how the acuracy?). Please add this information on the text.

Author Response

Dear Professor,

Thank you for helpful suggestions and comments, which have improved the quality of our manuscript greatly.

Revisions according to comments made by Reviewer 3:

The paper presents new concept for an all-fiber inclinometer based on a tapered fiber Bragg grating (tFBG) in fiber ring laser (FRL) with the capability of measuring the tilt angle and temperature, simultaneously. I have some suggestions to improve the paper.

Comment 1 Introduction section: it is missing the works related to FBG accelerometers and even FBGs based angle or tilt sensors, which is a large research field and there are many works to be reported. The focus of this paper is inclinometer and there is a lack of works cited about his concern. Please read and consider some references about this point in order to improve the introduction:

FBG acceleremeters: 1. Diaphragm Based Fiber Bragg Grating Acceleration Sensor with Temperature Compensation. Sensors 17, 218, 2017.

FBGs Angle/tilt measurements:

  1. A Rotation Independent In-Place Inclinometer/Tilt Sensor Based on Fiber Bragg Grating," IEEE Transactions on Instrumentation and Measurement, vol. 68, no. 8, pp. 2943-2953, Aug. 2019;
  2. Polymer optical fiber Bragg gratings in CYTOP fibers for angle measurement with dynamic compensation, Polymers 10 (6), 674, 2018; 4. FPI-POFBG Angular Movement Sensor Inscribed in CYTOP Fibers With Dynamic Angle Compensator, IEEE Sensors Journal 20 (11), 5962-5969, 2020.

Reply: Thanks for the reviewer’s comment. We add the corresponding literature in the introduction to further explain the development of inclinometer. [underlined part of the introduction].

Introduction part:

Different types of inclinometers and accelerometers based on broadband light sources have been reported [45-47].

Page 1

New references are being attached here

References for axial force measurements

  1. Li, T.; Tan, Y.; Han, X.;  Zheng, K.; Zhou, Z., Diaphragm Based Fiber Bragg Grating Acceleration Sensor with Temperature Compensation. Sensors (Basel) 2017, 17 (1).
  2. Maheshwari, M.; Yang, Y.; Upadrashta, D.; Chaturvedi, T., A Rotation Independent In-Place Inclinometer/Tilt Sensor Based on Fiber Bragg Grating. Ieee T Instrum Meas 2019, 68 (8), 2943-2953.
  3. Leal-Junior, A.; Theodosiou, A.; Diaz, C.;  Marques, C.;  Pontes, M. J.;  Kalli, K.; Frizera-Neto, A., Polymer Optical Fiber Bragg Gratings in CYTOP Fibers for Angle Measurement with Dynamic Compensation. Polymers (Basel) 2018, 10 (6).

Page 17

Comment 2. “The TFBG normally is used for Tilted Fiber Bragg Grating (TFBG) and it is not the best acronym for tapered FBG, maybe use tFBG.”

Reply: This is a really a thoughtful comment from the reviewer, we appreciate the reviewer comment.

We’ve change TFBG to tFBG accordingly.

Comment 3. “When the authors talk about the tilt angle and temperature measurements simultaneously, it must be more discussed and add more results because it is not well demonstrated as simultaneously way.”

Reply: We thank reviewer for his suggestions, now explanation has been described elaborately in the theoretical part of the paper. A new figure has been added to explain the results. A brief mechanism principle of the sensors is given in the introduction part.

To evaluate the temperature response of the inclinometer, the sensor was placed in a furnace with a temperature accuracy of 10°C, and a series of measurements at various inclination angles and within the range of 0°C–50°C was carried out. It was observed that the temperature changes resulted only in the wavelength shift of the spectrum, while the optical power level of the peak wavelength remained constant. As shown in figures the relative wavelength shift of the spectral response is a linear function of temperature and does not depend on the inclination angle. Thus, the temperature coefficient for the examined inclinometer was 12pm/°C which is a typical

value for Bragg gratings. Besides, the influence of temperature changes on the reflectivity is negligible. It was verified that even if the optical spectrum of the measurement setup was not perfectly flat within the wavelength range of the tFBG, a change in the temperature from 0°C to 50°C resulted in a variation of the optical power at the peak reflection wavelength of less than 0.15dBm, therefore was comparable with the tolerance of the set bending angle. Moreover, if the resultant spectral shape of the FRL and the circulator set are known and considered, the variation of the optical power level at peak wavelength can be reduced to a value much lower than the mentioned 0.1dBm. Due to the different spectral responses to changes in the temperature and bending angle, the tapered fiber Bragg grating in the configuration presented above allows for simultaneous measurement of inclination angle and temperature.

Comment 4. “More details about FBGs fabrication must be added to the manuscript. Nothing is added about FBG fabrication (please consider to add some literature about FBG fabrication in different materials such as Advances on polymer optical fiber gratings using a KrF pulsed laser system operating at 248 nm, Fibers 6 (1), 13, 2018.”

Reply: This is a really a thoughtful comment from the reviewer, we appreciate the reviewer comment.

Here, we add more analysis to explain the sensing mechanism and the usage of the two tapers/Bragg gratings. A mechanism principle of the sensors is given in the underline part.

Theoretical part:

The filter of the presented inclinometer is actuality a section of TFBG written on head face of the fiber. By heating and stretching, the taper was made on the standard single-mode matched FBG. Symmetrical structures with total length of 45 and 37 mm and waist diameter of 75 and 86 μm were obtained. In order to improve the photosensitivity of FBG, the TFBG cross section was hydrogenated at 120 bars for one week at room temperature. The growth and length of the TFBG reflectance spectrum were continuously monitored during the taper process.

The principle of operation of the presented inclinometer is based on two phenomena: bending losses and selective reflection. Light from a broadband optical source is launched into the optical fiber and then travels through the tapered section. When the fiber is bent, the mode field is shifted towards one side of the tapered fiber. Thus, a part of the optical power is coupled into the cladding region and finally partially dissipates. The remaining part of the radiation is selectively reflected by the fiber Bragg grating written into the transition of the taper. Then it travels again through the bending area and once again is partially lost. Finally, the output power spectrum is measured by an optical spectrum analyzer. The light outside the wavelength range of the grating passes through it towards the reflection-free termination. The main advantage of the TFBG application, as in most reflection based optical systems, is that the effect of the optical power loss occurs twice, which enhances inclinometer sensitivity. Moreover, the grating written in the taper transition ensures that the temperature can be simultaneously measured as close to the bending region as possible.

In this work, the taper profile chosen was a trade-off between ensuring the adiabaticity criterion and providing relatively high bending sensitivity of the analyzed inclinometer. If the radius of the tapered optical fiber changed too rapidly along its length, it would lead to shifting of the power of the LP01 mode to the higher order modes. Thus, when the waist diameter is too small, the taper is no longer adiabatic and becomes lossy. Moreover, to ensure reliable temperature measurements close to the bending region, some constraints on the maximum length of the taper and TFBG location are imposed. On the other hand, bending losses at a given inclination angle (and thus sensor sensitivity) increase with the reduction of the taper waist diameter. This may be concluded from the model presented by Marcuse  or from experimental data provided by Kim et al. It means that in contrast to the adiabaticity criterion, an optical fiber taper with adequately small waist diameter ensures satisfactory bending sensitivity of the inclinometer.

In conclusion, in order to ensure that the proposed configuration has enough performance, we choose the parameters of TFBG according to repeated experiments. Another way to measure the power spectrum of reflected light is to use an optical power meter instead of OSA. This can effectively reduce the cost of measurement.

Ref

[48] Marques, C.;  Leal-Junior, A.;  Min, R.;  Domingues, M.;  Leitão, C.;  Antunes, P.;  Ortega, B.; André, P., Advances on Polymer Optical Fiber Gratings Using a KrF Pulsed Laser System Operating at 248 nm. Fibers 2018, 6 (1).

Comment 5 “Figure 2. (a) Scheme of the normal FBG. (b) Scheme of the tapered FBG (86µm). (c) Scheme of the tapered FBG (75µm). For me, it is not a scheme but a microscope images right?

For a commercial application the instrumentation like OSA and SLS could be an issue due to portability and even expensive cost. Please, add some words about that and consider to include some literature about low-cost interrogation systems as a potential solution to decrease costs and increase the portability for a commercial solution. Please read and consider the literature: Fast peak-tracking method for FBG reflection spectrum and nonlinear error compensation," Opt. Lett. 45, 451-454 (2020); Wireless, Portable Fiber Bragg Grating Interrogation System Employing Optical Edge Filter. Sensors 19, 3222, 2019.”

Reply: Authors thank for the reviewer comments; it is very significant. it is not a scheme but a microscope images we’ve change the notes, accordingly.

In addition, the spectrometer is the most conventional detection instrument for detecting light intensity. We also choose to use the optical power meter for monitoring, and get the detection results consistent with the spectrometer measurement results. This verifies the stability and repeatability of the sensor. As an alternative instrument of spectrometer, optical power meter can greatly reduce the measurement cost. We further describe the consistency of the two instruments.

Figure.4.  optical power change of OSA and power meter

Ref:

  1. Wang, J.; Huang, T.; Duan, F.;  Cheng, Q.;  Zhang, F.; Qu, X., Fast peak-tracking method for FBG reflection spectrum and nonlinear error compensation. Optics Letters 2020, 45 (2).
  2. Ogawa, K.; Koyama, S.; Haseda, Y.;  Fujita, K.;  Ishizawa, H.; Fujimoto, K., Wireless, Portable Fiber Bragg Grating Interrogation System Employing Optical Edge Filter. Sensors (Basel) 2019, 19 (14).

Comment 6: “How about the repeatability of the proposed sensor (if carry out many cycles of tests) and about the reproducibility of the proposed sensor (due to tapering process we can obtain similar probes how the accuracy?). Please add this information on the text.”

Reply: We are thankful to the reviewers for his comments. In the point of repeatability and reproducibility, we have used the sensors for several times (>10 times) with change in measurands and find similar outcomes at each of the time. With computed value the error of the measurement is of the order of 10-3. The sensors were used for repeated experiments quite a few time (>10times and >5hours) and results were similar, so this sensor is repeatable. As for reproducibility, The filter of the presented inclinometer is actuality a section of TFBG written on head face of the fiber. By heating and stretching, the taper was made on the standard single-mode matched FBG. Besides, the TFBG cross section was hydrogenated at 120 bars for one week at room temperature. We get almost the same quality and shape of the tFBG.

We thank the reviewers once again for their helpful suggestions and comments, which have improved the quality of our manuscript greatly.

Yours sincerely,

Li-Yang Shao/Weihao Lin

Round 2

Reviewer 1 Report

The authors improved tha paper. It can now be published in the present form.

Reviewer 3 Report

The authors made sufficient improvements.